# Early Diagnosis of Intracranial Internal Carotid Artery Stenosis Using Extracranial Hemodynamic Indices from Carotid Doppler Ultrasound

**DOI:** 10.3390/bioengineering9090422

**Published:** 2022-08-29

**Authors:** Xiangdong Zhang, Dan Wu, Hongye Li, Yonghan Fang, Huahua Xiong, Ye Li

**Affiliations:** 1Shenzhen Institute of Advanced Technology, Chinese Academy of Sciences, Shenzhen 518055, China; 2Department of Mathematics, University of Macau, Macao 999078, China; 3Department of Ultrasound, The First Affiliated Hospital of Shenzhen University, Shenzhen Second People’s Hospital, Shenzhen 518035, China

**Keywords:** atherosclerosis, Doppler ultrasound, internal carotid artery, hemodynamic modeling, stroke

## Abstract

Atherosclerotic intracranial internal carotid artery stenosis (IICAS) is a leading cause of strokes. Due to the limitations of major cerebral imaging techniques, the early diagnosis of IICAS remains challenging. Clinical studies have revealed that arterial stenosis may have complicated effects on the blood flow’s velocity from a distance. Therefore, based on a patient-specific one-dimensional hemodynamic model, we quantitatively investigated the effects of IICAS on extracranial internal carotid artery (ICA) flow velocity waveforms to identify sensitive hemodynamic indices for IICAS diagnoses. Classical hemodynamic indices, including the peak systolic velocity (PSV), end-diastolic velocity (EDV), and resistive index (RI), were calculated on the basis of simulations with and without IICAS. In addition, the first harmonic ratio (FHR), which is defined as the ratio between the first harmonic amplitude and the sum of the amplitudes of the 1st–20th order harmonics, was proposed to evaluate flow waveform patterns. To investigate the diagnostic performance of the indices, we included 52 patients with mild-to-moderate IICAS (<70%) in a case–control study and considered 24 patients without stenosis as controls. The simulation analyses revealed that the existence of IICAS dramatically increased the FHR and decreased the PSV and EDV in the same patient. Statistical analyses showed that the average PSV, EDV, and RI were lower in the stenosis group than in the control group; however, there were no significant differences (*p* > 0.05) between the two groups, except for the PSV of the right ICA (*p* = 0.011). The FHR was significantly higher in the stenosis group than in the control group (*p* < 0.001), with superior diagnostic performance. Taken together, the FHR is a promising index for the early diagnosis of IICAS using carotid Doppler ultrasound methods.

## 1. Introduction

Atherosclerotic intracranial internal carotid artery stenosis (IICAS) is a leading cause of stroke across different races [1,2,3]. IICAS is normally diagnosed using cerebral digital subtraction angiography (DSA), computed tomography angiography (CTA), magnetic resonance angiography (MRA), or transcranial Doppler (TCD) ultrasound. DSA is the gold standard method for the quantitative evaluation of IICAS; however, this method is invasive and expensive [4]. CTA and MRA are less invasive than DSA; however, contrast agents are still needed, which may increase the risk of allergies and the deterioration of renal function [5]. Ultrasound is safer and less expensive; nevertheless, TCD ultrasound may have problems in locating arteries in some individuals [6]. Therefore, it is not widely applied compared with CTA/MRA/DSA. The early diagnosis of IICAS is crucial in preventing strokes and in reducing mortality. However, due to the disadvantages of these traditional methods, few asymptomatic patients in the early stage undergo these medical imaging examinations. The early diagnosis of IICAS thus remains challenging.

Because arterial stenosis may have complicated effects on blood flow velocities from a distance, correlations between invisible stenosis and hemodynamic indices measured using Doppler ultrasound, such as the peak systolic velocity (PSV), end-diastolic velocity (EDV), and resistive index (RI), have been widely investigated [7,8,9,10,11]. In addition, the Doppler spectrum waveform pattern may contain information on stenosis in other arterial locations [12]. Sakima et al. [13] divided left vertebral artery (VA) waveforms into five subtypes and found a significant correlation between the waveforms and the degree of left subclavian artery (SCA) stenosis. Chan et al. [14] proposed a new hemodynamic index (i.e., stenosis index (SI)) to quantitatively study the Doppler waveform patterns of the renal arteries. The index is calculated from the ratio between high- and low-frequency powers after applying the fast Fourier transform (FFT) to the waveforms. Their simulation results indicated that the SI may be a more effective diagnostic index for stenosis. In a subsequent study on the detection of significant transplant hepatic arterial stenosis, the SI outperformed the traditional RI and pulsatile index [15]. These studies indicate the possibility of detecting intracranial stenosis, which can only be imaged using CTA/MRA/DSA, from different arterial locations, such as the extracranial carotid arteries, using hemodynamic indices measured on ordinary Doppler ultrasound. Moreover, compared with CTA/MRA/DSA, Doppler ultrasound is a safer, low-cost, and easy-to-operate method that can be widely applied in physical examinations to facilitate the early diagnosis of IICAS.

Hemodynamic simulation is a powerful tool for quantitatively investigating the effects of stenosis on flow velocities and hemodynamic indices, which may facilitate the identification of effective hemodynamic indices for IICAS diagnosis. One-dimensional (1D) modeling of the arteries is a fast and effective modeling method for simulating pressure and flow-wave propagation in the human cardiovascular system [16]; it can fit Doppler ultrasound-measured flow waveforms well in actual patients [17]. Based on the 1D modeling of coronary arteries with stenosis, Yin et al. [18] developed a predictive probabilistic model of fractional flow reserve for coronary artery disease assessment. Their simulation analyses validated the efficiency of 1D models. Similar studies have validated the accuracy of 1D models with stenosis [19,20,21].

In this study, we developed a 1D patient-specific hemodynamic model and simulated extracranial internal carotid artery (ICA) waveforms with and without IICAS to quantitatively investigate the effects of IICAS on upstream ICA waveforms. Hemodynamic indices in the time and frequency domains were analyzed to identify sensitive indices for IICAS diagnoses. Two groups of patients with and without mild-to-moderate IICAS (<70%) were recruited to measure the Doppler waveforms at the extracranial segment of the ICA. Statistical analysis was performed to compare different hemodynamic indices, and their diagnostic performance was evaluated.

## 2. Materials and Methods

### 2.1. One-Dimensional Hemodynamic Model of the Human Cardiovascular System

In this study, we developed a patient-specific hemodynamic model of the cardiovascular system to simulate ICA blood flow velocity waveforms based on the validated modeling method in our previous work [17]. The model could simulate personalized dynamic variations in the blood pressure, flow velocity, and vessel diameter at every arterial location. As shown in Figure 1, the model consisted of a systemic artery tree (Figure 1A) combined with a cerebral artery network (Figure 1B). The governing equations for each point in each artery segment were the 1D incompressible viscous flow equations [17] coupled with the thick-wall linear elastic circular tube equation [22].
(1a)$$\frac{\partial }{\partial t}\left[\begin{array}{c} A\\ U \end{array}\right]+\frac{\partial }{\partial z}\left[\begin{array}{c} UA\\ \frac{U^{2}}{2}+\frac{P}{\rho } \end{array}\right]=\left[\begin{array}{c} 0\\ -K_{R}\frac{U}{A} \end{array}\right]$$
(1b)$$P-P_{0}=\frac{E(r_{1}^{2}-r_{0}^{2})}{1.5r_{1}^{2}}\cdot \frac{r-r_{0}}{r_{0}}$$

In the above equations, *A* is the artery cross-sectional luminal area; *r* is the inner radius of the artery (2π*r*^2^ = *A*); *P* is the corresponding pressure; *E* is the elastic modulus; *U* is the area-averaged flow velocity. *R*_0_ and *r*_0_ represent the inner and outer radii of the vessel when the pressure is *P*_0_, respectively (Figure 1C). The blood density is denoted by *ρ* and set at 1.06 g/cm^3^. *K_R_* is the friction force term and equals 8πν, assuming a parabolic flow velocity profile on the cross-sections [23], where ν denotes the dynamic viscosity and is set to 4.43 s^−1^cm^2^.

At the inlet of the ascending aorta (segment no. 1 in Figure 1A), a prescribed volumetric flow rate curve was modeled as the boundary condition (Figure 1D), and the area under the curve was equal to the stroke volume. At the distal ends of the arteries, widely used three-element RCR Windkessel models were used to model peripheral vessels (Figure 1E). More details regarding the modeling methods and numerical schemes can be found in the article by Zhang et al. [17]. The default parameters of each artery segment, including the diameter, wall thickness, and elastic modulus, were taken from the literature [24,25].

### 2.2. Patient-Specific Hemodynamic Modeling

The major parameters of the cardiovascular system model were the stroke volume, heart rate, artery diameter, artery wall elasticity, peripheral resistance, and peripheral compliance. To develop a patient-specific model, we recruited patients from the Second People’s Hospital in Shenzhen, China. MRA showed that the patients had a single stenosis (43%) in the left intracranial ICA and an intact Willis circle. The stroke volume and heart rate were measured noninvasively using B-mode and M-mode ultrasound methods. B-mode ultrasound was used to measure the diameters of the ICA (nos. 40 and 47 in Figure 1A), common carotid artery (nos. 5 and 11), external carotid artery (nos. 39 and 48), VA (nos. 6 and 16), brachiocephalic artery (no. 3), and SCA (nos. 4 and 15). We also measured the diameters at several locations of the aorta, including the ascending aorta (no. 1), aortic arch (no. 10), thoracic aorta (no. 12), and abdominal aorta (no. 31), and the remaining diameters were determined via linear scaling. The geometry of the left IICAS, including the length and degree of stenosis, was estimated from MRA images. The elastic modulus, peripheral resistance, and compliance of the relevant vessels were tuned automatically to match the simulated ICA waveforms with the measured waveforms based on the Levenberg–Marquardt optimization algorithm. The measured and tuned parameters are shown in the Appendix A. The measurement procedures were approved by the Institutional Review Board of the Second People’s Hospital of Shenzhen. The entire procedure was explained to the patients, and written consent was obtained. More details regarding the personalized modeling method can be found in the article by Zhang et al. [17].

To study the effects of IICAS on upstream hemodynamic indices, we compared the simulation results of the patient-specific model in two different cases—with and without a developed IICAS. In the normal case, we removed the geometrical narrowing in the model and reduced the local vessel’s wall stiffness by 50% because increased local carotid stiffness is believed to be associated with the presence of atherosclerosis [26], while the other parameters of the cardiovascular system model remained unchanged. The PSV, EDV, and RI of proximal ICA flows were calculated for each case based on the simulated ICA blood-flow velocity waveforms. The RI was calculated using the following formula: RI = (PSV-EDV)/PSV.

In addition to the typical hemodynamic indices in the time domain, features in the frequency domain were also considered. The amplitudes of each harmonic frequency were obtained by applying the FFT to a single-period digitalized flow waveform. According to Chan et al. [14], high-frequency waves may be dampened in stenotic vessels. Therefore, in this study, we propose a new index named the first harmonic ratio (FHR), which is the ratio between the first harmonic amplitude and the sum of amplitudes from the 1st to the 20th order in order to investigate the possible high-frequency damping effect:(2)$$\mathrm{FHR}=\frac{AMP_{1}}{\sum_{i=1}^{20}AMP_{i}}$$
where *AMP_i_* denotes the amplitude of the *i*th order harmonic. We set the maximum order to 20 because most features of the frequency spectrum can be included in this frequency range, and the amplitudes of the higher orders are close to zero. Theoretically, when high-order harmonics are dampened, the FHR should be elevated.

### 2.3. Measurement of the Hemodynamic Indices in the Patient Groups

We identified patients who underwent both carotid ultrasound and cerebral CTA/MRA or DSA at the Second People’s Hospital of Shenzhen (in 1 month) from 1 January 2019 to 31 December 2020. The cohort was divided into two groups: with and without mild-to-moderate IICAS (<70%). The degree of stenosis was calculated using the North American Symptomatic Carotid Endarterectomy Trial criteria [27], in which 70% is the cutoff value between severe and moderate stenosis. The following patients were excluded from the study: patients with severe stenosis, extracranial or intracranial stenosis at other locations, or heart or kidney disease or those who had undergone cardiovascular surgery. Finally, 52 and 24 patients were included in the stenosis and control groups, respectively. In the stenosis group, 14 patients had a single stenosis in the left ICA; 12 patients had a single stenosis in the right ICA; 26 patients had stenosis on both sides. There were no significant differences in the average age, sex ratio, hypertension rate, and hyperlipidemia rate between the two groups. However, the difference in the diabetes rate was relatively significant (*p* < 0.05), probably because diabetes is a major risk factor for atherosclerotic stenosis [28]. Detailed information on the two groups is presented in Table 1.

Left and right ICA flow waveforms were measured using the linear array ultrasound transducer L12-3 (3–9 MHz) of Philips EPiQ-7C. The measuring point was located at the distal segment of the extracranial ICA within the detectable range (Figure 2A). IICAS was identified from the CTA/MRA/DSA images (Figure 2B). The PSV, EDV, and RI were recorded (Figure 2C). The original Doppler waveform images containing envelop curves were saved to obtain digitalized waveform data (Figure 2D), and the amplitudes of the different orders in the frequency domain were acquired by applying the FFT to the digitalized velocity waveforms (Figure 2E). Thereafter, the index FHR in the frequency domain was calculated for each patient. This study was approved by the Institutional Review Board of the Second People’s Hospital of Shenzhen.

### 2.4. Statistical Analysis

To determine the differences between the hemodynamic indices of the two groups, we calculated the average value ± standard deviation of the PSV, EDV, RI, and FHR. Differences between the two groups were quantitatively evaluated using *t*-tests. Multivariate regressions were performed to investigate multiple risk factors for IICAS diagnosis and multiple contributing factors for hemodynamic index variations. Receiver operating characteristic (ROC) curves of sensitive index were analyzed, and the area under the ROC curve (AUC) value was calculated for each index to evaluate diagnostic performance. For an index with a high AUC value (AUC value of >0.8), the optimized critical value was calculated on the basis of the maximum Youden index [29].

## 3. Results

### 3.1. Patient-Specific Simulation of the ICA Waveforms

The simulated left ICA waveforms of the personalized hemodynamic model were compared to the measured waveforms in Figure 3B. The total converged mean squared error between the measured and simulated waveforms was 3.5 (cm/s)^2^. Based on the figure, the 1D artery network model fits the Doppler waveforms well with similar major features, which indicates that the parameters are properly individualized. During the diastolic period, the measured Doppler waveforms showed small fluctuations, which may have been caused by turbulence, vortices, or measurement errors.

We compared the simulated ICA waveforms of patients with and without IICAS while the other system parameters remained unchanged, as shown in Figure 3 and Table 2. Based on the figure, the stenosis will decrease both the PSV (−12.6 cm/s) and EDV (−6.1 cm/s), with a slightly lower amplitude (PSV-EDV) in the stenosis case. The RI remained unchanged (0.53). When the stenosis was removed, the FHR decreased dramatically from 0.390 to 0.280. Meanwhile, the simulated ICA waveform without stenosis showed typical features of ordinary individuals, with a steep upstroke and the following high platform. Therefore, we infer that IICAS tends to decrease the PSV and EDV slightly while increasing the FHR significantly. Thus, the FHR may be a promising index for IICAS diagnosis.

The mechanism of FHR elevation in the presence of developed stenosis may result from the characteristics of wave propagation in the arteries. In a fluid-filled elastic tube, wave propagations occur not only in the fluid but also in the elastic wall [30]. Because of the viscous effect, waves in the blood flow are dominated by low-frequency waves, whereas elastic waves are dominated by high-frequency reflected waves. When the wall thickness or stiffness increases, the amplitude of the elastic waves decays due to the reduced radius changes, which may lead to FHR elevations.

### 3.2. Diagnostic Performance of the Hemodynamic Indices

The left and right ICA flow waveforms with and without IICAS were analyzed separately. As shown in Table 3, the average PSV, EDV, and RI were lower in the stenosis group than in the control group; however, there was no significant difference between the EDV and RI on both sides (*p* > 0.05). The average right PSV in the stenosis group was significantly different from that in the control group (*p* < 0.05); however, the difference in the left PSV between the two groups was lower (*p* = 0.067). All AUC values of the PSV, EDV, and RI were below 0.8, indicating poor performance in stenosis diagnosis.

In contrast to the classical indices, the FHR showed a superior diagnostic performance. The t-test results showed a significant difference between the two groups on both sides (*p* < 0.001). The AUC values of the left and right sides were 0.838 and 0.836, respectively. The best cutoff value for the left FHR was 0.363, with a sensitivity of 70% and a specificity of 91.7%. The best cutoff value for the right FHR was 0.351, with a sensitivity of 76.3% and a specificity of 79.2%.

The FHR of both sides is plotted as a scatter plot in Figure 4A, where the labeled IDs correspond to the IDs in the Appendix A. Based on the figure, most negative cases are located in the lower left quarter, separated from the positive cases. A relatively high maximum FHR on both sides usually indicates stenosis in the left or right intracranial ICA. Therefore, a more accurate diagnosis with higher sensitivity and specificity may be realized by considering measurements from both sides, regardless of the stenosis location. Therefore, we conducted a statistical analysis to determine whether the maximum FHR can distinguish patients with and without IICAS on either or both sides. The results demonstrated a significant difference between the two groups (*p* < 0.001). The AUC value was 0.888, which was higher than that of the single-side diagnosis. The best cutoff value was 0.360, with a sensitivity of 88.5% and a specificity of 83.3%, as shown by the ROC curve in Figure 4B. Multivariate logistic regression was further performed to investigate possible confounding effects of age, gender, and basic diseases. As shown in Table 4, diabetes is a significant risk factor (*p* < 0.05) for IICAS, and the FHR remains significant after considering multiple factors.

### 3.3. Multiple Contributing Factors for Hemodynamic Index Variations

The simulation results indicated that the existence of IICAS will lead to significant FHR elevations, and we inferred that the mechanism of FHR elevations is caused by increased artery wall stiffness. In addition to the development of stenosis, other factors may also contribute to FHR variations and have some effects on IICAS diagnosis. Multivariate linear regression was performed to investigate the relations between the FHR and multiple factors. As shown in Table 5, stenosis remains the leading cause of FHR elevations. Moreover, gender also has significant effects on the FHR, with females tending to have higher FHR than males. Therefore, we inferred that females tend to have higher arterial stiffness and more severe atherosclerotic stenosis, which is consistent with recent studies [31,32,33].

In addition to the above factors, medications may also lead to hemodynamic index variations. We collected information on drug use in 50 subjects in this study and performed multivariate regressions to investigate the impact of medications. The drugs were classified into five categories: (1) calcium channel blockers (CCBs, e.g., Amlodipine); (2) cerebral vasodilators (e.g., Betahistine mesylate); (3) angiotensin receptor blockers (ARBs, e.g., Valsartan) and other vasodilators; (4) hypoglycemic drugs (e.g., Metformin); (5) hypolipidemic drugs (e.g., Atorvastatin). The vasodilators were classified into three sub-categories based on different hemodynamic effects: CCBs may decrease the arterial stiffness of large arteries by blocking calcium ions into smooth muscle cells; cerebral vasodilators may reduce cerebral vascular resistance; ARBs and other vasodilators may decrease system vascular resistance. Table 6 demonstrates that the maximum FHR is insensitive to medications.

Regressions about other hemodynamic indices reveal that the left PSV is significantly affected by the usage of cerebral vasodilators, the EDV is affected by age and hyperlipidemia, and the RI is affected by age. Details of the regression results can be found in the Appendix A.

## 4. Discussion

In this study, we quantitatively investigated the effects of mild-to-moderate IICAS on proximal ICA flow waveforms using hemodynamic simulations and statistical analyses in a group of patients. The pattern of the entire waveform was quantitatively evaluated using the FHR proposed in this study, which is the ratio between the amplitudes of the low- and high-frequency waves. A 1D patient-specific hemodynamic model was developed to simulate ICA waveforms with and without stenoses. Based on the results, the removal of the stenosis will lead to an increase in the PSV and EDV and a dramatic decrease in the FHR. Statistical analysis was performed on actual patient groups to test the diagnostic performance of these indices. We found that the patients with IICAS tended to have a lower PSV, lower EDV, and significantly higher FHR than those without IICAS. Moreover, the FHR showed a good performance in IICAS diagnosis, with a sensitivity of 88.8% and a specificity of 83.3%, during dual-side carotid ultrasounds.

The diagnostic performances of the classical hemodynamic indices (PSV, EDV, and RI) in this study are poor compared with those reported in previous studies [8,9,10] in which severe stenosis is usually considered, and the distance between the stenosis and probe location is close despite some of the indices, such as the PSV, showing differences between the two groups to some extent. The probable cause of this phenomenon is that the classical indices may not be sensitive to mild-to-moderate stenosis from a relatively long distance, which was considered in this study. In addition, the simulated results with stenosis had a lower EDV (−6.1 cm/s) than the difference between the averaged group values (−3.1 cm/s). Hence, we infer that the compensatory vasodilation of the peripheral vessels [34,35] may also occur in narrowing the differences, because the parameters of the peripheral vessels remain unchanged in the simulation, while compensatory peripheral resistance reduction may occur in actual patients as stenosis develops.

In contrast to the classical indices, which usually use one or two single values in the Doppler waveforms, the FHR proposed in this study use information on the entire waveform, potentially making the index more sensitive to small variations induced by mild-to-moderate stenosis. The diagnosis of mild-to-moderate stenosis is more clinically significant than that of severe stenosis because severe stenosis is usually accompanied by symptoms, whereas early diagnosis and therapy of IICAS can effectively prevent severe diseases, such as strokes. Moreover, multivariate regressions reveal that the classical indices may be easily influenced by age, hyperlipidemia, and cerebral vasodilators, while the FHR is insensitive to age, basic diseases, and drug usage.

The AUC value of the single-sided FHR was 0.838 for the left ICA and 0.836 for the right ICA, while the AUC value for the maximum FHR obtained from both sides was 0.888; this finding indicates that FHR-based diagnosis is more accurate in distinguishing patients with or without IICAS than in identifying the stenosis location. Accordingly, the FHR is a promising diagnostic index for the early diagnosis of IICAS. This diagnostic method can be applied in ordinary physical examinations to identify possible patients with IICAS in the early stages, and cerebral CTA/MRA can be further applied to locate the stenosis. In addition, FHR-based diagnosis may be effective in other arteries where it is difficult to locate the stenosis directly using ultrasound images.

A personalized hemodynamic model was used to investigate the effects of IICAS on the complicated cardiovascular system. The simulation analyses demonstrated that the 1D artery network model was capable of simulating Doppler ultrasound waveforms precisely, making it a potentially useful tool in other Doppler ultrasound-related studies. In addition, personalized cardiovascular function assessment is possible by solving the inverse problem of identifying parameters from the measured Doppler waveforms. For example, the risk of atherosclerotic stenosis may be evaluated using the quantitative analysis of vascular stiffness at different arterial sites.

## 5. Limitations

In addition to age, gender, basic diseases and medications collected in this study, other contributing factors may also affect the results. The abnormal bending of ICA (or “Dolichocarotids”) is a significant risk factor for cardiovascular events [36], and artery curvatures may have complicated effects on blood flow. However, because cerebral images usually focus on intracranial regions, parts of ICA segments are missing in many images. Scopes of the cerebral images should be adjusted to obtain complete ICA segments in future studies, and the effects of Dolichocarotids should be investigated.

A total of 76 participants were included in this study; however, the sample size was not large enough to test the diagnostic efficacy of the FHR, and the sizes of the two groups were not well balanced because the positive cases outnumbered the negative cases in the hospital. Multi-center studies should be conducted in the future to expand the sample’s size. Another limitation of the statistical study is the lack of gold standard cerebral images; only two patients were diagnosed using DSA in this study because CTA/MRA is less invasive and expensive. One of the cases had different diagnostic conclusions from CTA and DSA, as shown in Figure 5. Due to a locally insufficient contrast agent, IICAS was misdiagnosed in the CTA scan. In contrast, the gold standard DSA showed no stenosis in that region. Interestingly, the FHR of the patient was within the normal range (below 0.360 on both sides), which was in accordance with the DSA diagnosis. If there are more similar cases in the positive group, the actual diagnostic performance of the FHR may be of higher quality. Therefore, more data labeled with DSA images should be collected in the future.

## 6. Conclusions

The effects of IICAS on the extracranial hemodynamic indices were quantitatively investigated using a 1D patient-specific hemodynamic model of the human cardiovascular system. A significant dampening of high-order harmonics was found in the extracranial ICA flow waveforms in the presence of IICAS. Therefore, we proposed a new index called the FHR to quantitatively evaluate this effect. Using carotid Doppler ultrasound measurements, we further conducted a case–control study including 76 patients; we found that the FHR had a superior diagnostic performance for mild-to-moderate IICAS (<70%) and that the classical indices showed no significant differences between the stenosis and control groups. Multivariate regressions revealed that the classical indices were susceptible to age, hyperlipidemia, and cerebral vasodilators, while IICAS remained the dominant factor for FHR elevations. FHR measurements using carotid Doppler ultrasound may facilitate the early diagnosis of IICAS.

## Figures and Tables

**Figure 1 bioengineering-09-00422-f001:**
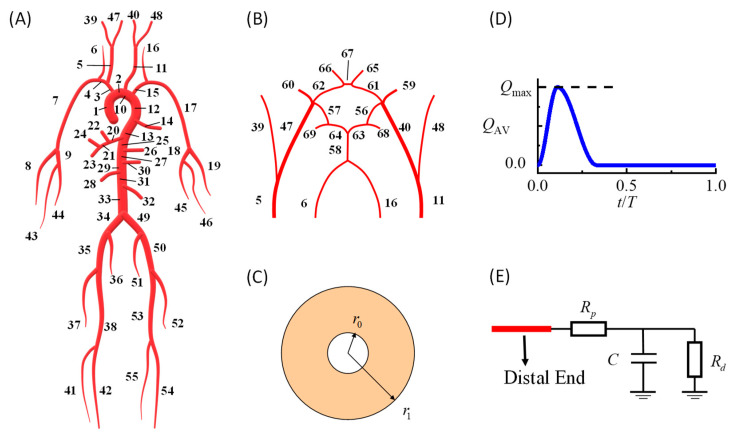
Hemodynamic simulation model of the human cardiovascular system: (**A**) 1D model of the systemic artery tree; (**B**) 1D model of the Willis circle; (**C**) cross-section of the artery model; (**D**) prescribed flow rate curve at the inlet of the ascending aorta; (**E**) peripheral vessel model at the outlets of the artery networks; 1D, one-dimensional.

**Figure 2 bioengineering-09-00422-f002:**
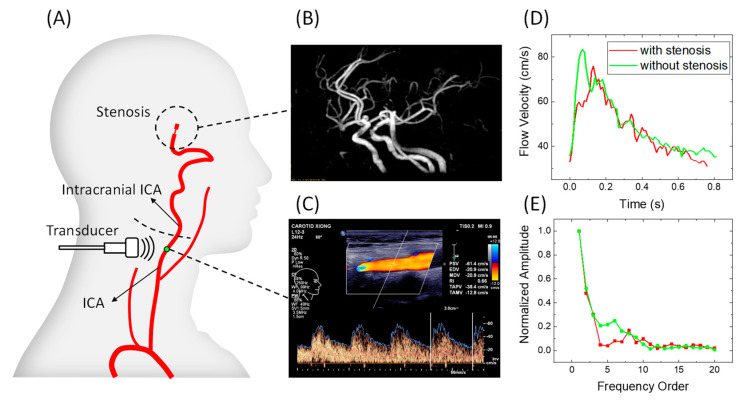
Schematic images of the measurement procedure: (**A**) schematic diagram of a stenosis location and the Doppler ultrasound measurement location; (**B**) magnetic resonance angiography image of a patient with IICAS; (**C**) original image of ICA flow waveforms and measurements of the PSV, EDV, and RI; (**D**) typical digitalized single-period flow velocity waveforms of a patient with IICAS and a patient without IICAS; (**E**) normalized harmonic amplitudes of waveforms in (**D**). ICA, internal carotid artery; IICAS, intracranial internal carotid artery stenosis; PSV, peak systolic velocity; EDV, end-diastolic velocity; RI, resistive index.

**Figure 3 bioengineering-09-00422-f003:**
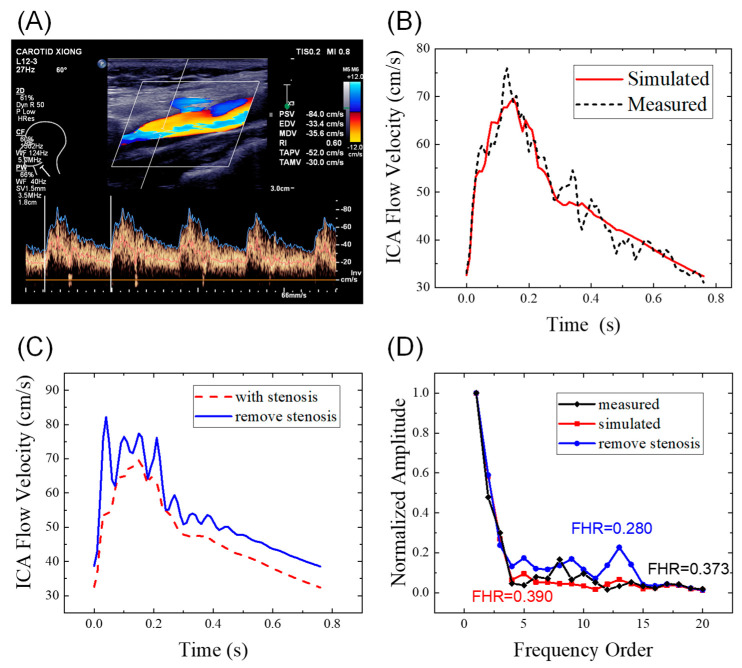
ICA flow waveforms of a patient: (**A**) original Doppler ultrasound data; (**B**) comparison of the measured data and simulated results from the hemodynamic model; (**C**) comparison of the simulated waveforms with and without stenoses; (**D**) comparison in the frequency domain. ICA, internal carotid artery; FHR, first harmonic ratio. The patient ID is 18 in the Appendix A.

**Figure 4 bioengineering-09-00422-f004:**
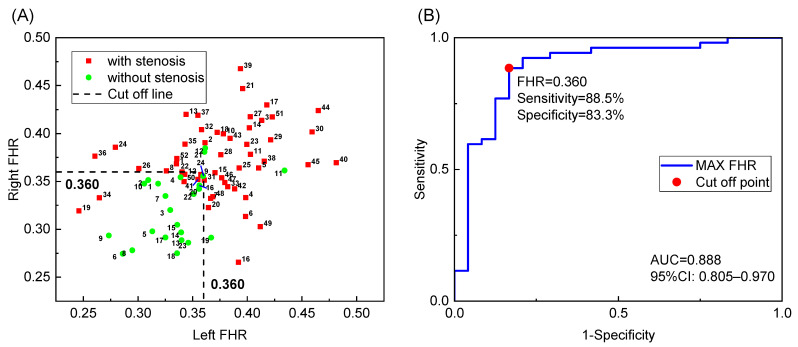
Diagnosis of the patients with and without IICAS: (**A**) FHR data of both sides; (**B**) receiver operating characteristic curve of IICAS diagnosis using the maximum FHR of both sides. FHR, first harmonic ratio; IICAS, intracranial internal carotid artery stenosis; AUC, area under the receiver operating characteristic curve; CI, confidence interval.

**Figure 5 bioengineering-09-00422-f005:**
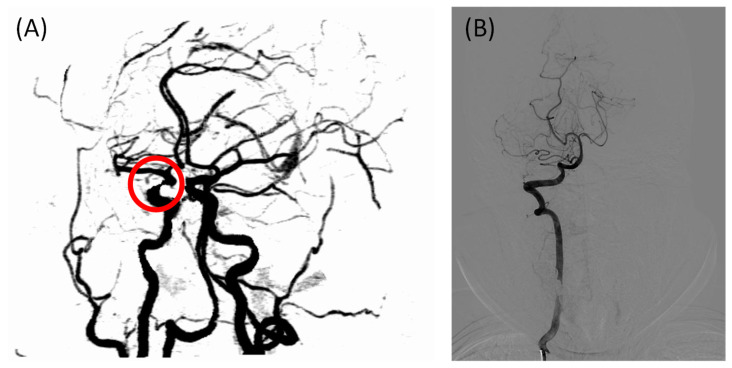
Incorrectly labeled case in the CTA scan due to an insufficient contrast agent, with the FHR in the normal range: (**A**) CTA image of the intracranial arteries of the patient, with the misdiagnosed intracranial internal carotid artery stenosis indicated in the red circle; (**B**) gold standard intracranial digital subtraction angiography image of the same patient showing no stenosis in the arteries, which is consistent with the FHR-based diagnosis. CTA, computed tomography angiography; FHR, first harmonic ratio.

**Table 1 bioengineering-09-00422-t001:** Participant information.

	Stenosis	Normal	*p* Value
Total Number	52	24	-
Male Gender	29 (56%)	14 (58%)	0.837
Hypertension	32 (62%)	11 (46%)	0.204
Diabetes	20 (38%)	3 (13%)	0.022
Hyperlipidemia	17 (33%)	5 (21%)	0.296
Age	65.4 ± 7.9	62.8 ± 8.7	0.210

**Table 2 bioengineering-09-00422-t002:** Hemodynamic indices of a patient.

	PSV	EDV	RI	FHR
Measured	75.9	33.0	0.57	0.373
Simulated (with stenosis)	69.6	32.6	0.53	0.390
Simulated (without stenosis)	82.2	38.7	0.53	0.280

**Table 3 bioengineering-09-00422-t003:** Comparison of the hemodynamic indices between the patients with and without intracranial internal carotid artery stenosis.

	Stenosis	Control	*p* Value	AUC	95%CI
N (left)	40	24			
L. PSV	72.4 ± 19.9	82.7 ± 23.4	0.067	0.637	0.495–0.779
L. EDV	25.4 ± 8.1	28.5 ± 10.1	0.180	0.580	0.437–0.722
L. RI	0.642 ± 0.086	0.653 ± 0.087	0.598	0.566	0.420–0.711
L. FHR	0.380 ± 0.045	0.336 ± 0.033	<0.001	0.838	0.721–0.954
N (right)	38	24			
R. PSV	67.9 ± 15.6	78.9 ± 16.8	0.011	0.696	0.556–0.837
R. EDV	26.2 ± 8.9	27.1 ± 9.3	0.712	0.521	0.375–0.668
R. RI	0.612 ± 0.102	0.653 ± 0.102	0.132	0.605	0.463–0.747
R. FHR	0.372 ± 0.038	0.323 ± 0.035	<0.001	0.836	0.729–0.942

**Table 4 bioengineering-09-00422-t004:** Results of the logistic regression (N = 76).

	Coefficient	Standard Error	*p* Value	Odds Ratio
intercept	−28.052	7.810	<0.001	-
age	0.012	0.040	0.763	1.012
100× max FHR	0.726	0.197	<0.001	2.066
male gender	1.128	0.802	0.160	3.088
hypertension	0.345	0.748	0.644	1.413
diabetes	1.948	0.947	0.040	7.016
hyperlipidemia	0.668	0.840	0.426	1.951

**Table 5 bioengineering-09-00422-t005:** Relations between the maximum First Harmonic Ratio and multiple factors (N = 76).

	Coefficient	Standard Error	*p* Value
Intercept	0.3285	0.0302	<0.001
Age	0.0004	0.0005	0.371
Male Gender	−0.0204	0.0078	0.011
Hypertension	0.0002	0.0081	0.980
Diabetes	−0.0017	0.0088	0.852
Hyperlipidemia	0.0033	0.0087	0.704
Stenosis	0.0487	0.0086	<0.001

**Table 6 bioengineering-09-00422-t006:** Relations between the maximum First Harmonic Ratio and medications (N = 50).

	Coefficient	Standard Error	*p* Value
Intercept	0.3435	0.0145	<0.001
CCBs	0.0142	0.0113	0.214
Cerebral Vasodilators	0.0032	0.0132	0.811
ARBs et al.	0.0118	0.0118	0.321
Hypoglycemic Drugs	−0.0040	0.0143	0.781
Hypolipidemic Drugs	0.0081	0.0107	0.454
Stenosis	0.0528	0.0123	<0.001
Male Gender	−0.0210	0.0105	0.053

## Data Availability

All the data involved in this study can be found in Appendix A.

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
