# Peer review of "Early Diagnosis of Intracranial Internal Carotid Artery Stenosis Using Extracranial Hemodynamic Indices from Carotid Doppler Ultrasound"

_bioengineering, 2022, doi:10.3390/bioengineering9090422_

Round 1
Reviewer 1 Report
This is an interesting study. I would like to recommend its publication. I have several minor comments:
First, is this a clinical study? If yes, please provide the detail of IRB approval.
Second, I would like the author to clarify its study type in abstract session.
Third, the abbreviations used in Figure 2 should be given in thier full spelling in the legend portion. The same rule should be applied for other figures.
Author Response
Comments to the Author:
This is an interesting study. I would like to recommend its publication. I have several minor comments:
Answer:
We thank the referee for the positive evaluation and the recommendation.
First, is this a clinical study? If yes, please provide the detail of IRB approval.
Answer:
This study is a combination of a numerical simulation study and a clinical study. The IRB approval was stated in the “Institutional Review Board Statement” section at the end of the manuscript. To emphasize the approval, we have added the statements in the main text (section 2.2 and 2.3, Page 4 and 5).
Second, I would like the author to clarify its study type in abstract session.
Answer:
We have revised the abstract to point out that the test in patient groups is a kind of case-control study (Page1).
Third, the abbreviations used in Figure 2 should be given in thier full spelling in the legend portion. The same rule should be applied for other figures.
Answer:
Full spellings of the abbreviations are now used in each figure caption.
Reviewer 2 Report
To:
Editorial Board
Bioengineering
Title: “Early diagnosis of intracranial internal carotid artery stenosis using extracranial hemodynamic indexes from carotid Doppler ultrasound”
Dear Editor,
I evaluated this paper and I think that:
- The English of the paper should be revised due to typos. Please provide.
- What about the impact of dolichocarotids on these analyses? Authors should discuss such a point in relation to the paper from Ciccone MM et al. J Atheroscler Thromb. 2014;21(1):56-63.
- Why did the authors use 70% as the cut off for carotid stenosis? Please deeply specify this point.
- What about the medications of patients? These might impact on results. Please describe medications and discuss such a point.
- A multivariate regression analysis should be performed in order to evaluate the impact of confounding factors on final results.
Author Response
Comments to the Author:
I evaluated this paper and I think that:
The English of the paper should be revised due to typos. Please provide.
Answer:
We have revised the entire manuscript according to suggestions from native speakers.
What about the impact of dolichocarotids on these analyses? Authors should discuss such a point in relation to the paper from Ciccone MM et al. J Atheroscler Thromb. 2014;21(1):56-63.
Answer:
We thank the referee for providing the useful information. Dolichocarotids (DCs) are likely to be a risk factor for atherosclerotic stenosis, and artery curvatures may have complicated effects on hemodynamics. However, we found only parts of ICA were shown in many medical images, and we couldn’t identify whether these patients have DCs. We have listed this issue in the limitation section and we will collect data with complete ICA images in future studies. (Page 10)
Why did the authors use 70% as the cut off for carotid stenosis? Please deeply specify this point.
Answer:
70% is the cutoff value between severe stenosis and moderate stenosis according to the North American Symptomatic Carotid Endarterectomy Trial (NASCET) criteria. Because the aim of this study is early diagnosis of IICAS, only mild to moderate stenosis (<70%) was considered. We have revised the first paragraph in section 2.3 to clarify the criterion and its origin. (Page 4)
What about the medications of patients? These might impact on results. Please describe medications and discuss such a point.
Answer:
We thank the referee for the insightful suggestion. We have got drug records from 50 participants. Based on different effects, we have classified the drugs in use into 5 categories: (1) calcium channel blockers (CCBs); (2) cerebral vasodilators; (3) angiotensin receptor blockers (ARBs) and other vasodilators; (4) hypoglycemic drugs; (5) hypolipidemic drugs. After multivariate linear regression analysis, we found cerebral vasodilators may have significant effects on Peak Systolic Velocity (PSV), while the FHR index is insensitive to drug treatments. We have added medication-related content in this article and the supplementary data. (Page 9)
A multivariate regression analysis should be performed in order to evaluate the impact of confounding factors on final results.
Answer:
Logistic multivariate regression has been performed to investigate possible confounding effects of age, gender, and basic diseases. We found diabetes is also a significant risk factor for IICAS. The FHR remains significant (p<0.001) after considering confounding factors. We have also set the hemodynamic indices as outcomes and analyzed contributing factors for index variations using multivariate linear regressions. According to the analysis, the FHR is also affected by gender, while classical indices PSV, EDV, and RI may be sensitive to age, hyperlipidemia, and usage of cerebral vasodilators. Overall, FHR is a robust and accurate index for IICAS diagnosis. We have largely revised the manuscript to demonstrate these new results. (Page 8-9)
Reviewer 3 Report
In their study, the authors took up a significant problem from the clinical point of view: the intracranial internal carotid artery (IICA) atherosclerotic stenosis being one of the leading causes of stroke. I agree with researchers that, due to the shortcomings of significant brain imaging techniques, an early diagnosis of IICA is still tricky, as arterial stenosis can have a complex effect on blood flow velocities at a distance. An interesting proposal of the researchers is to propose the first harmonic coefficient (FHR) to assess the flow patterns, which was defined as the ratio between the first harmonic amplitude and the sum of the amplitudes of 1st-20th order harmonics. To test the diagnostic performance of investigator-assessed indexes 52 patients with mild to moderate IICA (<70%) were enrolled in this study, and 24 patients without stenosis were grouped into a control group.
The simulation studies showed that the existence of IICA significantly increases the FHR and reduces the PSV and EDV for the same person. The statistical results of the patient groups showed that the mean PSV, EDV, and RI values ​​are lower in the stenosis group. However, for the most part, there were no statistically significant differences, except for the PSV of the right IICA between the two groups, indicating that these measures may not be helpful. On the other hand, FHR turned out to be a promising indicator of the early diagnosis of IICA using cervical Doppler ultrasound.
In general, I believe that the study was adequately planned and carried out. The introduction of the article is an appropriate introduction to the subject. I think that it contains all the necessary information. The methodology is correct and well presented, and I do not think it requires any correction. Statistical analyzes are accurate, and their selection is valid.
Results - their presentation is correct. The figures are legible and, in my opinion, are not manipulative. The discussion correctly relates to the results obtained.
My comments / questions
1. Why is there a difference between the total number in the stenosis group (n = 52) in Table 1 and the stenosis group in Table 3? In Table 3, you can see that the left side is 40 and the right side is 38 - does this mean that some people had stenosis on the L and P sides, and therefore, the total number of stenosis analyzed is 52? Is that how it should be understood? In the discussion, the authors wrote (that a total number of 76 patients were included in this study). Should it be assumed in this situation that almost everyone, except for two persons, has stenosis on both sides? I checked it in the supplementary file and still don't understand these numbers.
By the way, I have the question of which stenosis was taken into account; I mean, which value was the lowest taken into consideration?
2. I find the superscripts a-d in Tables 2 and 3 redundant. Just an explanation under the table of abbreviations used. As a rule, such indices are used to indicate some statistical dependencies, and there are no such indices here.
3. Did the authors assess whether there is an influence of gender on the analyzed parameters? Especially I mean the FHR, but also the others. Of course, I am aware of the small group size, and it seems that extracting such information may be a problem. It is similar to the influence of diabetes and hypertension on the assessed parameters. Especially in this context, I was thinking about the reference group, called by researchers "normal," in which diabetes significantly differentiates between two groups. I mean normal and stenosis, which may affect the results.
4. Is it possible to indicate in Figures 3 and 5 which patients from the xls file were taken into account?
5. Have the patients from the xls file ID 8, 11 (negative file) and ID 26, 27, 36, 37 (positive file) numbers been included in table 1?
I mean, the percentage of patients with diabetes and hypertension? It turns out that it is not; hence the information under the table that there was no data from several people.
6. I have a suggestion to extract some passages from the discussion and insert them into the Limitation of the study. It would be more legible.
7. There are no conclusions in the article. Could the authors briefly summarize the results of their research?
I think the article is exciting and could be accepted for possible publication after the abovementioned issues have been clarified.
Author Response
We very much appreciate the insightful summary and the positive evaluation by the referee.
My comments / questions
- Why is there a difference between the total number in the stenosis group (n = 52) in Table 1 and the stenosis group in Table 3? In Table 3, you can see that the left side is 40 and the right side is 38 - does this mean that some people had stenosis on the L and P sides, and therefore, the total number of stenosis analyzed is 52? Is that how it should be understood? In the discussion, the authors wrote (that a total number of 76 patients were included in this study). Should it be assumed in this situation that almost everyone, except for two persons, has stenosis on both sides? I checked it in the supplementary file and still don't understand these numbers.
By the way, I have the question of which stenosis was taken into account; I mean, which value was the lowest taken into consideration?
Answer:
The difference is caused by the fact that some patients have stenosis on both the left and right sides. The number of patients who have left IICAS only is 14, the number of patients who have right IICAS only is 12, and the number of patients with IICAS on both sides is 26, as indicated in section 2.3 (Page 4). Therefore, the number of left ICA with stenosis is 14+26=40 and 12+26=38 for the other side. The numbers in Table 3 are in fact artery numbers rather than participant numbers (one person has two ICAs), and the control group in Table 3 only includes patients without IICAS on either side. The statement “76 patients” in the Discussion means the total number of the subjects including 52 participants with IICAS and 24 participants without IICAS. We used the word “patients” because all data were collected from the hospital, and most of the subjects had some kind of diseases. To avoid misunderstandings, we have replaced the word “patients” with “participants” in the discussion. (Page 10)
Therefore, all stenoses were taken into account. As mentioned above, some patients have one stenosis and others have two stenoses. The total number of stenoses is 78 (38+40), and they are distributed among 52 patients.
- I find the superscripts a-d in Tables 2 and 3 redundant. Just an explanation under the table of abbreviations used. As a rule, such indices are used to indicate some statistical dependencies, and there are no such indices here.
Answer:
We have deleted the redundant superscripts in Table 2 and 3.
- Did the authors assess whether there is an influence of gender on the analyzed parameters? Especially I mean the FHR, but also the others. Of course, I am aware of the small group size, and it seems that extracting such information may be a problem. It is similar to the influence of diabetes and hypertension on the assessed parameters. Especially in this context, I was thinking about the reference group, called by researchers "normal," in which diabetes significantly differentiates between two groups. I mean normal and stenosis, which may affect the results.
Answer:
We thank the referee for the insightful suggestion. We have performed multivariate regressions to investigate the influence of gender and other factors. Interestingly, female subjects tend to have higher FHR values than male subjects. According to some recent studies, we infer that this phenomenon is caused by higher artery stiffness (larger elastic modulus or thicker artery walls) in females, and females tend to develop more severe stenosis. We have also analyzed contributing factors for the PSV, EDV, and RI variations, and the results indicated that age, hyperlipidemia, and usage of cerebral vasodilators may have significant impacts on these classical indices. Therefore, the FHR is also a more robust index for its insensitivity to most factors. Logistic regression was also performed to investigate confounding factors for IICAS diagnosis, and we found diabetes is also a significant risk factor for IICAS. However, the FHR remains significant (p<0.001) after considering confounding factors. We have largely revised the manuscript to demonstrate these new results. (Page8-9)
- Is it possible to indicate in Figures 3 and 5 which patients from the xls file were taken into account?
Answer:
We have added small patient IDs near the scatter points in Figure 5. The patient in Figure 3 was also included in the case-control study, and we have mentioned his ID in the caption. (Page 6 and 8)
- Have the patients from the xls file ID 8, 11 (negative file) and ID 26, 27, 36, 37 (positive file) numbers been included in table 1?
I mean, the percentage of patients with diabetes and hypertension? It turns out that it is not; hence the information under the table that there was no data from several people.
Answer:
They are labeled as negative (0) and included in table 1. We have re-confirmed the data source with the hospital, all subjects have medical records. For those who have “/” symbols in the supplementary data, doctors just neglected the description of their disease history. Therefore, we simply assumed that they didn’t have the basic diseases. We have added descriptions under the supplementary tables.
- I have a suggestion to extract some passages from the discussion and insert them into the Limitation of the study. It would be more legible.
Answer:
We have separated the discussion and added a new “limitation” section. (Page 10)
- There are no conclusions in the article. Could the authors briefly summarize the results of their research?
Answer:
Yes, we have added a conclusion section to summarize our major findings. (Page 11)
I think the article is exciting and could be accepted for possible publication after the abovementioned issues have been clarified.
Answer:
We thank the referee again for the insightful suggestions.